# The Role of the Hyperfine Structure for the Determination of Improved Level Energies of Ta II, Pr II and La II

**Laurentius Windholz**

Institute of Experimental Physics, Graz University of Technology, Petersgasse 16, A-8010 Graz, Austria; windholz@tugraz.at

**Abstract:** For the determination of improved energy levels of ionic spectra of elements with large values of nuclear magnetic dipole moment (and eventually large values of nuclear quadrupole moments), it is necessary to determine the center of gravity of spectral lines from resolved hyperfine structure patterns appearing in highly resolved spectra. This is demonstrated on spectral lines of Ta II, Pr II and La II. Blend situations (different transitions with accidentally nearly the same wave number difference between the combining levels) must also be considered.

**Keywords:** energy levels; Ta II; Pr II; La II

## 1. Introduction

The laser spectroscopy group at Graz University of Technology has been concerned since 1990 with investigations of the hyperfine (hf) structure of several elements. The spectra of tantalum, praseodymuim and lanthanum were investigated most intensely. As a source of free atoms, a hollow cathode lamp was used in which a low-pressure plasma of the treated element was generated by cathode sputtering. For starting the discharge, a noble gas (argon or neon) at a typical pressure of 0.5 mbar was used. This source of free atoms and ions was investigated by tunable laser light (band width ca. 1 MHz) by scanning the laser frequency across the selected wavelength range. Either laser-induced fluorescence light or the change of the discharge impedance (optogalvanic detection) was observed. Details of the experimental arrangement can be found in various publications, e.g., [1–3].

In this paper, spectra of Ta, Pr and La are treated. These elements have in their natural abundance either only one dominant isotope (Ta, La, see Table 1) or are isotopically pure (Pr). Their nuclear magnetic dipole moment μ is large enough to cause hyperfine splitting of the spectral lines larger than the Doppler width in the spectra. Thus, in most cases, the observed hf structure can be used as valuable help for the classification of the spectral lines. The isotope composition and nuclear moments can be found in Table 1. For Pr and La, the quadrupole moment is quite small and can be neglected for most of the energy levels.

**Table 1.** Isotope composition and nuclear moments of the investigated elements (natural abundance). In the spectra of Ta and La, we observed only the dominant isotopes [181]Ta and [139]La.

| Element | Z | Isotope | Natural Abundance % | Lifetime (Years) | Nuclear Spin Quantum Number I | Magnetic Moment μ (μ$_N$) | Electric Quadrupole Moment Q ($10^{-28}$ m$^2$) |
|---|---|---|---|---|---|---|---|
| Ta | 73 | 180 | 0.012 | $1.2 \times 10^{15}$ | 9 | +4.825(11) | +4.95(2) |
| Ta | 73 | 181 | 99.998 | stable | 7/2 | +2.3705(7) | +3.28(6) |
| Pr | 59 | 141 | 100 | stable | 5/2 | +4.2754(5) | −0.059(4) |
| La | 57 | 138 | 0.09 | $1.05 \times 10^{11}$ | 5 | +3.713646(7) | +0.45(2) |
| La | 57 | 139 | 99.91 | stable | 7/2 | +2.7830455(9) | +0.20(1) |

\* Based on uncorrected proton moment, 2.79277564 nm. Values of μ and Q from [4].

While at the early stage of the investigations, the hf constants of already known energy levels were determined, and it turned out later that the list of energy levels given in literature [5,6] is far from being complete. Thus, the focus was directed to the finding of new energy levels in order to explain spectral lines that could not be classified as transitions between known energy levels. An overview of how previously unknown energy levels can be found is given in Ref. [7].

In order to get accurate start wavelengths for laser spectroscopic investigations, spectra with high resolution and high wavelength precision are needed. These requirements can be fulfilled by means of Fourier-transform (FT) spectroscopy. Several co-operations led to the availability of spectra of Ta [8], Pr [9], and La [10]. In these spectra, much more lines can be found than listed in commonly used wavelength tables [11]. The spectra were taken with a resolution between 0.03 and 0.05 cm$^{-1}$ and carefully wavelength calibrated using Ar II lines [12].

For strong lines which were classified and for which the hf constants of the combining levels were known, one finds that the center of gravity (cg) wavelengths determined from the FT spectra usually differ from wavelengths calculated from the known level energies. The conversion from wave numbers to standard air wavelengths and back was performed using the formula given by Reeder and Peck [13] for the refractive index of air.

Thus, exploiting the low uncertainty of the cg wavelengths determined from the FT spectra, improved level energies were determined. This was made step by step, beginning with the upper levels combining with the ground level. From these upper levels, transitions to lower levels were searched and energies of low-lying levels were corrected, and so on. Finally, the level energies were determined by a global fit procedure.

Since each spectrum contains several tenths of thousands of spectral lines, and since the treated elements have several hundreds or thousands energy levels, one can imagine that this procedure is very time-consuming. Thus, first the spectra of the first ions, Ta II, Pr II and La II, were used to perform a final determination of level energies. For most of the levels, an uncertainty of the level energy below 0.01 cm$^{-1}$ was achieved.

For the determination of the hf constants of the levels involved in an investigated transition (hf resolved spectra either from an FT spectrum or a laser spectroscopic scan), we used a software called "Fitter" which was very helpful [14].

It is clear that a suitable computer program is needed to manage such huge numbers of lines and the extended FT spectra. Thus, a program called "Elements" was developed (for descriptions, see Refs. [7,15]). One can select a certain wavelength and then go from one line to the next. The corresponding part of the FT spectrum is automatically shown. For classified lines, the combining levels and the hf pattern is shown in graphical form. For unclassified lines, classification suggestions (transitions between known energy levels within a selected wave number deviation) are also shown. A part of the FT spectrum can be copied easily to a simulation window where it can be compared with such a suggestion, for which the hf pattern is graphically shown. If no agreement between the pattern from the FT spectrum and any suggestion can be found, one has to assume that a previously unknown energy level is involved in the structure.

In the following sections, peculiarities of the investigated spectra are discussed in more details.

## 2. Ta II

Energy levels of Ta II are listed in the famous tables of Moore [5]. The given data are based mainly on works of Kiess [16], who published separately a collection of Ta II energy levels including the classified lines. Concerning the hf structure, a relatively low number of publications can be found. In 1952, Brown and Tamboulian [17] determined for the first time the nuclear moments of $^{181}$Ta investigating the hf structure of 7 Ta II lines. In 1987, Engleman Jr. [18] determined the hf constants of several Ta II levels and improved the energy values, but the results were presented only at a symposium but never published. Eriksson et al. [19] investigated in 2002 the Ta II spectrum with respect to applications in astrophysics. Laser spectroscopic determinations of the hf constants of Ta II were performed by Messnarz and Guthöhrlein [20,21] at approximately the same time. During work on his thesis, Messnarz discovered some new energy levels of Ta II and could correct some incorrect classifications. Zilio and Pickering [22] investigated an FT spectrum of Ta II and published hf constants of several levels.

The FT spectrum taken by J. Pickering at Imperial College London and spectra taken at the Kitt Peak Observatory by Engleman Jr. were used later in Graz [8,23,24]. In these papers, the discovery of new energy levels of Ta I and the determination of hf constants of already known Ta I levels are reported. The papers are part of a series of works published in Zeitschrift für Physik, the succeeding European Journal of Physics and later in Physica Scripta, entitled "Investigation of the hyperfine structure of Ta I lines, Part I to X". As can be seen from the list of authors, a strong collaboration between the group in Graz and the group of G. H. Guthöhrlein in Hamburg took place.

In the early stage of the investigations, it was very helpful to have a tool for distinguishing Ta I and Ta II spectral lines. This could be done with the help of photographic spectra, taken with a classical spectrograph, using for one trace of the spectrum a direct current (DC) hollow cathode lamp and, for a second trace, a discharge with pulsed excitation. In the pulsed discharge, the ionic lines are much more intense, allowing a clear distinguishing between Ta I and Ta II lines. One example of such spectra—in comparison with the FT spectra—is given in Ref. [25]. Another example where two spectral lines, one belonging to Ta I and the second to Ta II, are located side by side (Figure 1).

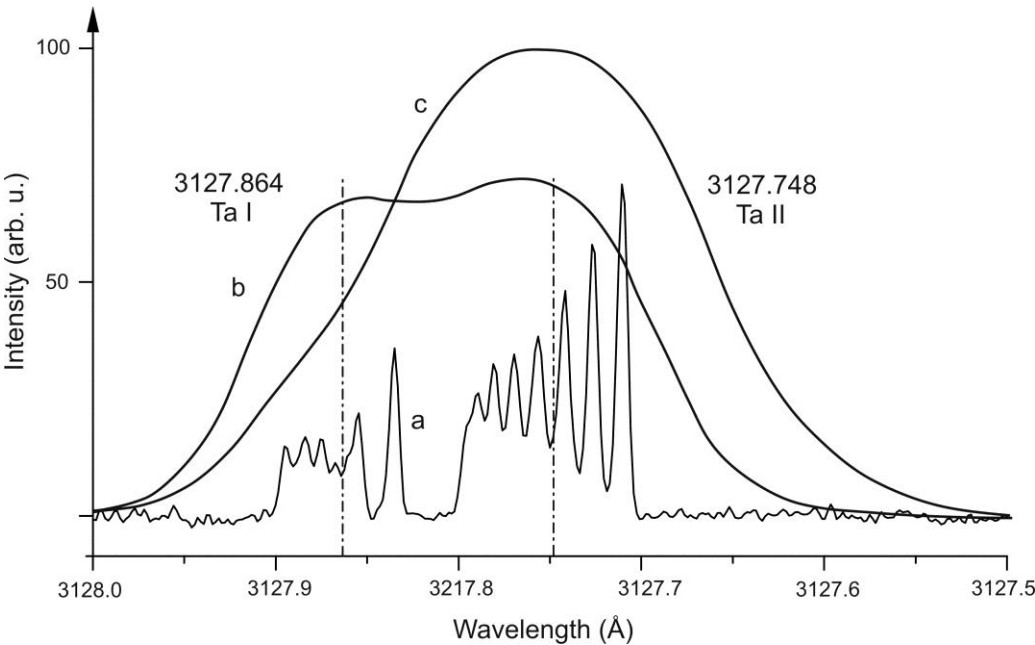

**Figure 1.** Comparison of a part of a highly resolved Fourier-transform (FT) spectrum with photographic spectra. For a description, see text.

In Figure 1, trace a shows the FT spectrum, full width at half maximum (FWHM) ca. 2.2 GHz. The light source was a hollow cathode lamp, operated by DC. The hf patterns of the lines are well resolved. The cg positions are marked with vertical dash-dotted lines. Traces b and c show photographic spectra, digitized from a photo plate generated by means of an Ebert-mounted grating spectrograph, focal length 2 m, 7th order (dispersion 0.72 Å/mm). The resolution is ca. 25 GHz (0.08 Å). In trace b, a hollow cathode lamp operated by DC was used, while, in trace c, the discharge was pulsed. This pulsed operation enhanced significantly the intensity of the ionic line compared to the atomic one. Thus, this spectral line on the photo plate causes difficulties in finding the cg wavelengths, since width and line center position strongly depend on the ratio of the intensities of the two lines and thus from the discharge conditions. Nevertheless, the two different photographic spectra made it easy to identify the structure at 3127.748 Å as an ionic line. The classification of the lines (and also of the lines in the subsequent figures) is given in Table A1 (see Appendix A).

In the FT spectra from Kitt Peak Observatory, an electrodeless microwave discharge was used as the light source. In these spectra, one can find a large number of unclassified lines. Some of them showed well resolved hf patterns. Analyzing these lines, a previously unknown system of high lying even parity ionic states with energies above 72,000 cm$^{-1}$ could be found, while the highest previously known even level is located at 40,900 cm$^{-1}$. First, results were published (together with the description of the classification program "Elements") in 2002 [15], all new energy levels in Ref. [26].

For finding accurate cg wavelengths of lines, it is very important to take into account their hf pattern. Figure 2 shows an example where, in the FT spectrum, practically only a single peak is visible. However, treating the peak wavelength as cg is not correct. Components with small intensities, but located far from the highest peak, shift the cg wavelength to the middle of the low-frequency wing of the peak.

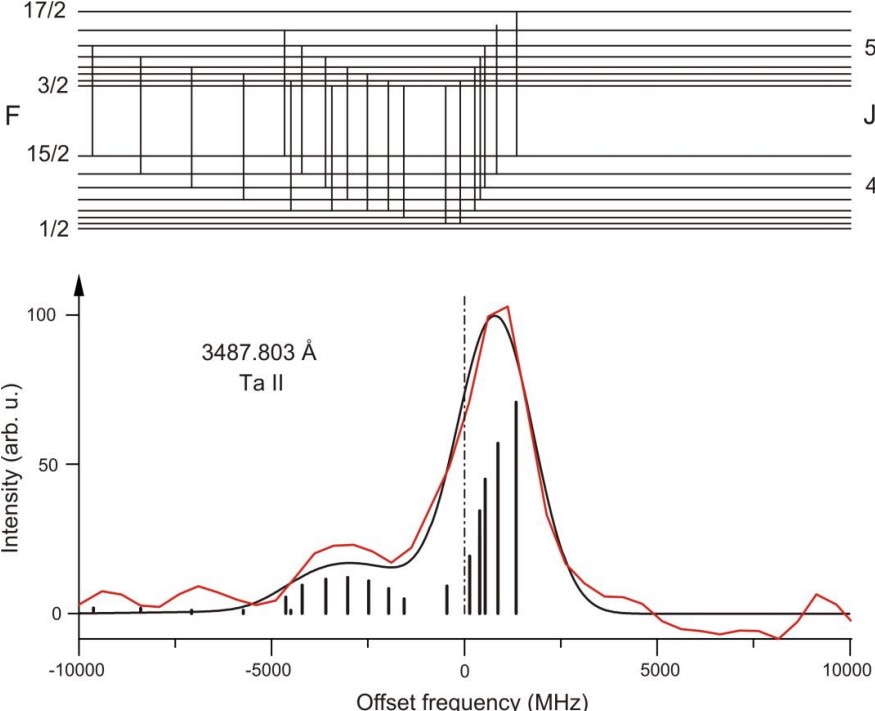

**Figure 2.** Example of a Ta II spectral line where the center of gravity (cg) wavelength is different from the single large peak appearing in the FT spectrum. Red line: FT spectrum, black line: simulation, full width at half maximum (FWHM) 2.2 GHz. Shown is also the hyperfine (hf) level scheme, the transitions and the components (theoretical intensity ratios). The components on the left side of the high peak (built by overlapping hf components for which ΔF = ΔJ) are only barely visible in the FT spectrum. The cg is shifted against the peak by 0.8 GHz (0.05 Å). The cg wavelength is marked by a vertical line (chain-dotted).

During the procedure of line classification and cg wavelength determination, one is quite often confronted with blend situations. As an example, Figure 3 shows a blend situation of two Ta II lines. With known hf constants of the four involved energy levels, it is possible to decompose the observed structure into two overlapping lines and to determine their cg wavelengths (in this case differing by 0.029 Å).

In the Ta FT spectra, ranging from 2120 Å to 46,000 Å, one can find a number of 12,200 spectral lines, from which as many lines as possible were classified. Around 1000 of them are Ar I or Ar II lines, which can be used for wavelength calibration. Ca. 3000 lines belong to the Ta II spectrum. Despite of all efforts, roughly 2000 lines are still not classified.

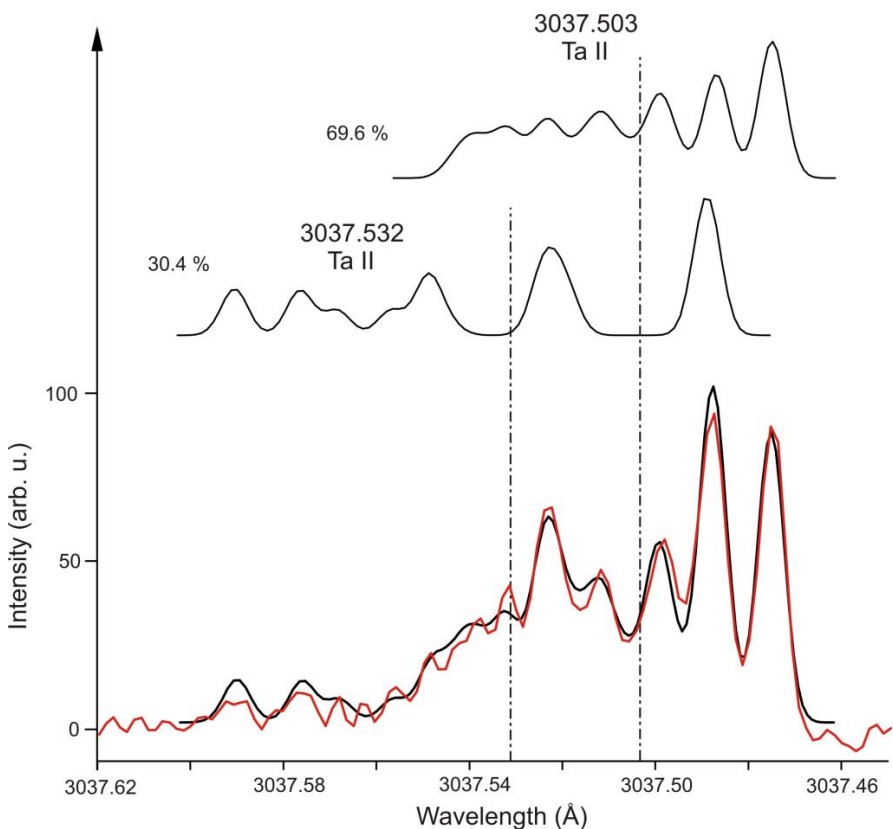

**Figure 3.** A blend situation of two Ta II lines. Red line: FT spectrum, black line: simulation, FWHM 2.2 GHz. In the upper part, normalized hf spectra of both lines are shown. Both profiles, added with the percentage given in the figure, gave the simulated sum profile, which describes the observed structure quite well. Thus, both cg wavelengths can be determined with high accuracy from the FT spectrum.

A systematic investigation of Ta II lines in the FT spectra available in Graz was performed in the PhD work of Uddin [27]. The determination of the hf constants of Ta II levels, either from laser spectroscopic records of Ta II lines or from the hf patterns of Ta II lines in the FT spectrum, made it possible to determine quite accurate values of cg wavelengths. From the classified Ta II lines, lines with good signal-to-noise ratio (SNR) were selected in order to re-calculate the level energies. Using the obtained vacuum wave numbers, a transition matrix was built up, and, in a global least squares fit, values for the level energies were calculated, together with their statistical uncertainties. The result was published recently [25].

The even parity system of Ta II was investigated theoretically—based on the results given in ref. [25]—by Stachowska et al. [28] performing a semi-empirical analysis, which confirmed also the new high lying levels above 72,000 cm$^{-1}$ (Figure 4). For this analysis, an important point was to exclude energy levels given in literature but in reality not existing. This is sometimes very difficult,

especially if it could not be verified under which assumptions (which lines) the corresponding level was introduced.

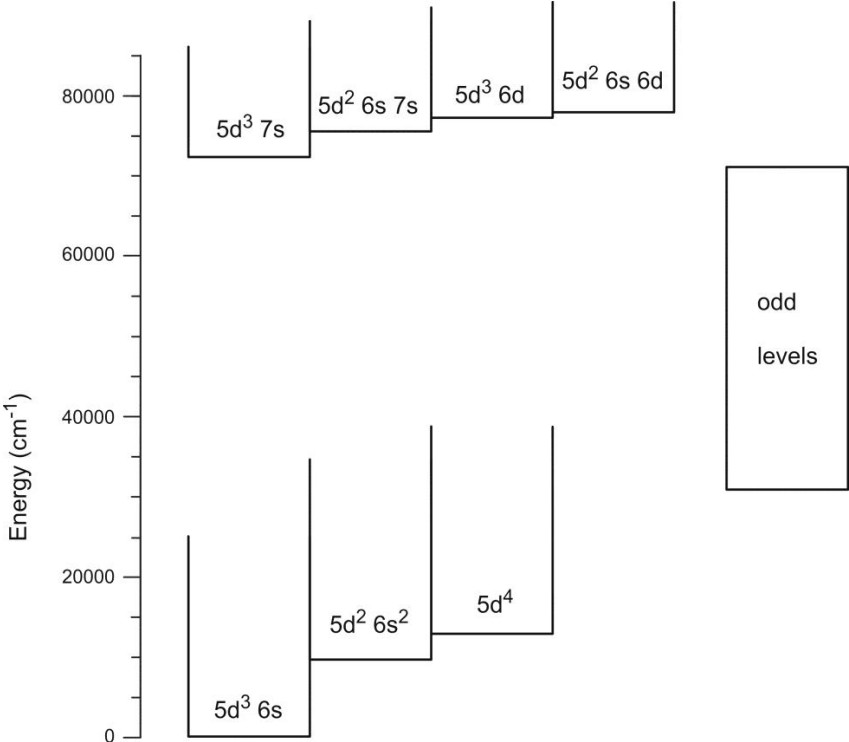

**Figure 4.** Simplified level scheme of Ta II. The theoretical analysis showed that the new even parity energy levels above 72,000 cm$^{-1}$ belong to the configurations shown in the picture.

## 3. Pr II

First investigations of the hf structure of Pr II lines were performed in 1929 by White [29]. In 1941, Rosen et al. [30] investigated Zeeman patterns of Pr II lines and determined Landé factors and J values of 74 energy levels. Further progress in finding fine structure levels was made by Blaise et al. [31] in 1973. All available data on Pr energy levels (and of all other atoms of the lanthanide group) were collected by Martin et al. in 1978 [6]. The work of Blaise was continued by Ginibre [32–35], who achieved remarkable progress in the classification of Pr I and Pr II lines. She discovered a large number of previously unknown energy levels and determined their hf constants. In 2001, Ivarsson et al. [36] improved the wavelength accuracy of some Pr II lines of astrophysical relevance. Investigations of the hf structure and search for new energy levels were performed also by the groups in Hamburg [37] and Graz. First results were published in a common paper [38], but still many of the results achieved in Hamburg are still not published.

The available FT spectra of Pr (3260–9880 Å) were taken by members of the group in Graz using an FT spectrometer in Hannover (group of Prof. Tiemann) and a hollow cathode lamp brought from Graz to Hannover. A first analysis of the spectra revealed more than 9000 previously unknown spectral lines of Pr I and Pr II, from which ca. 1200 could be classified as transitions between already known energy levels. During this first examination, 24 previously unknown energy levels were also discovered [9]. Later, these FT spectra were very helpful for laser spectroscopic investigations since the excitation wavelength could be set precisely to interesting peaks in the FT spectrum.

In between the list of spectral lines in the FT spectrum ca. 30,000 lines are contained, among them only 200 Ar lines and 650 Pr II lines. All other lines belong to the spectrum of neutral Pr (Pr I). In some spectral regions, the number of lines is so big that nearly no wavelength can be found at which the Pr plasma does not emit light. For example, the region around 5800 Å is shown in Figure 5.

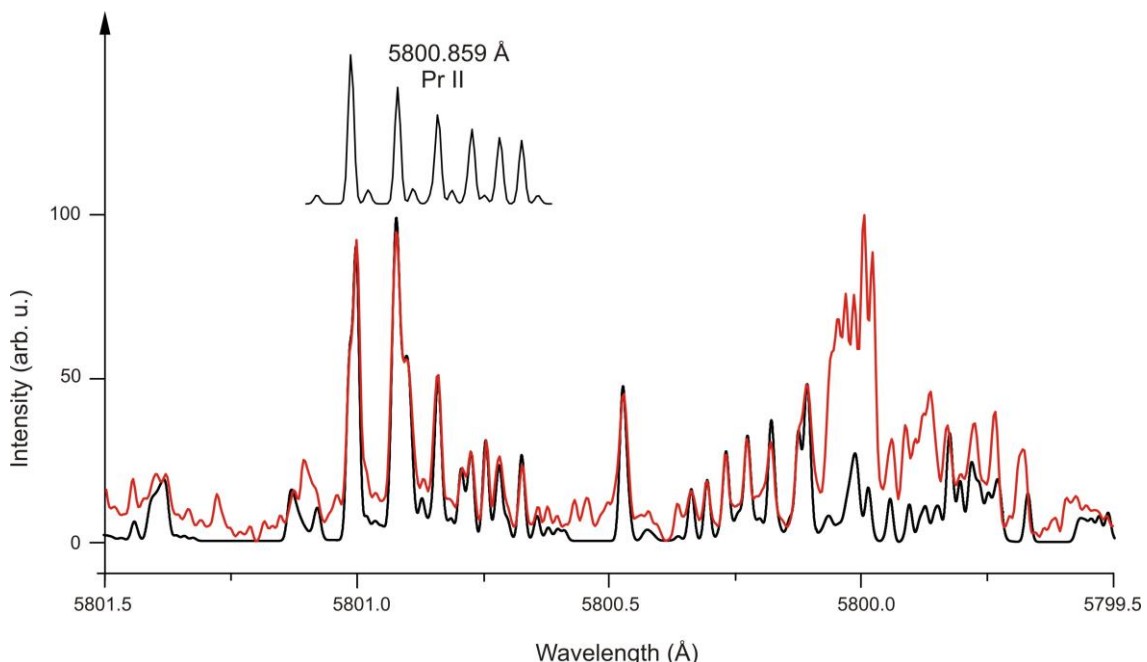

**Figure 5.** Part of an FT spectrum of Pr, using a direct current (DC) hollow cathode as light source (red line). As can be seen, there is nearly no wavelength at which the Pr plasma emits no light. The black curve shows a simulation taking into account all classified lines (FWHM 1.2 GHz). As can be seen (especially around 5800.0 Å), there are some quite dominant structures that are still not interpreted as transitions between known energy levels. Among the spectral lines, there is also one Pr II line (hf pattern shown in the upper part of the figure).

An overview concerning the already published newly discovered Pr I energy levels can be found in Ref. [39] and references therein.

In contrary to Ta II, even and odd levels of Pr II are not separated by a large energy difference. Thus, for the determination of improved energy values, one first has to build up two transition matrixes separately. These two could then connected by only one line, 4048.132. Even though this line had a low SNR of only 6, it must be used. A simplified level scheme is shown in Figure 6.

The FT spectra were also used to improve the accuracy of Pr II level energies. As can be seen from Figure 5, it may sometimes be tricky to find Pr II lines among the manifold of Pr I lines. Fortunately, in the blue and near infrared region, the identification becomes easier. Nevertheless, blend situations are quite frequently observed. As an example, in Figure 7, a blend of three lines is shown. In a more noisy spectrum or a spectrum with less resolution, one would notice only the dominant peak of the Pr I line at 5161.717 Å. Decomposing the observed pattern also allows for a precise determination of the cg wavelength of the involved Pr II line.

In Figure 8, a blend of two Pr II lines having nearly the same cg wavelength is shown. The wavelength difference is only 0.0025 Å (0.454 GHz or 0.015 cm$^{-1}$). The observed structure is well reproduced by adding the profiles of the two Pr II lines.

The wavelength of selected Pr II lines were determined carefully and a global fit of the level energies was performed. The results are published in Ref. [40].

Beside laser spectroscopic experiments, in which the plasma of a hollow cathode discharge was used as source of free Pr atoms, highly precise investigations of the hyperfine structure of Pr II lines were also made using collinear laser-ion beam spectroscopy (CLIBS) [41]. Such experiments were also performed earlier by Rivest et al. [42]. During the last time, CLIBS investigations were performed in the presence of a magnetic field in order to re-determine the Landé $g_J$-factors of the involved levels [43,44].

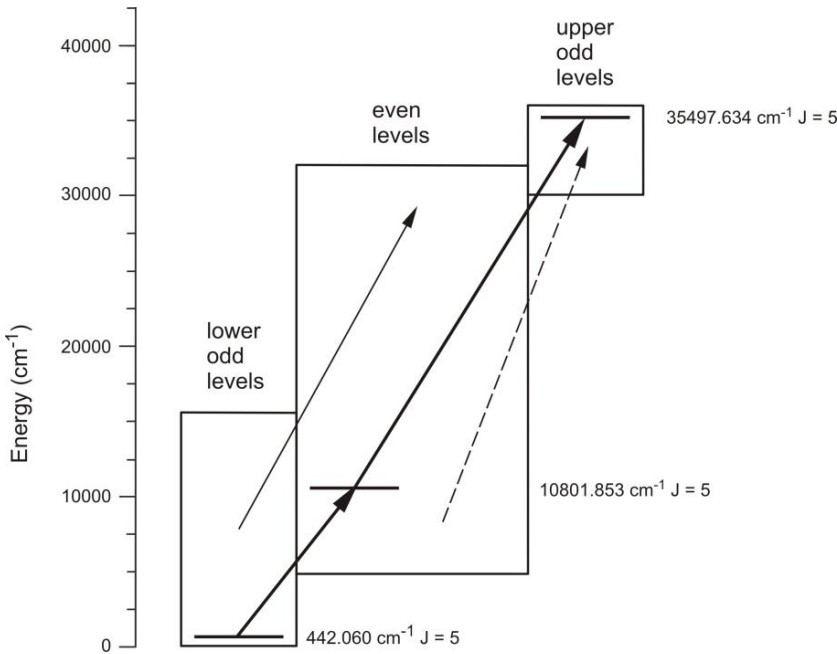

**Figure 6.** Simplified level scheme of Pr II. All transitions from lower odd levels to high-lying even levels are symbolized by the thin full arrow, all from the even levels to the upper odd levels by the dashed arrow. Only one ladder (bold arrows) could be found, which allowed for combining the two sub-systems of transitions.

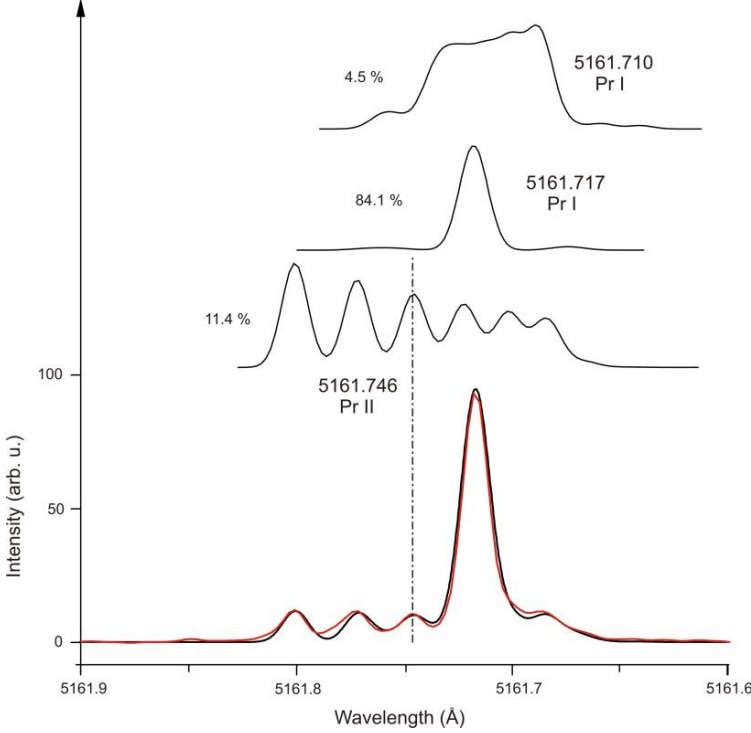

**Figure 7.** Blend of a Pr II line and two Pr I lines. Red line: FT spectrum, black line: simulation, FWHM 1.6 GHz). Knowing the hf constants of the involved levels, the cg wavelengths of all lines can be determined. In a less resolved spectrum, only the strong peak of the Pr I line at 5161.717 Å would be noticeable.

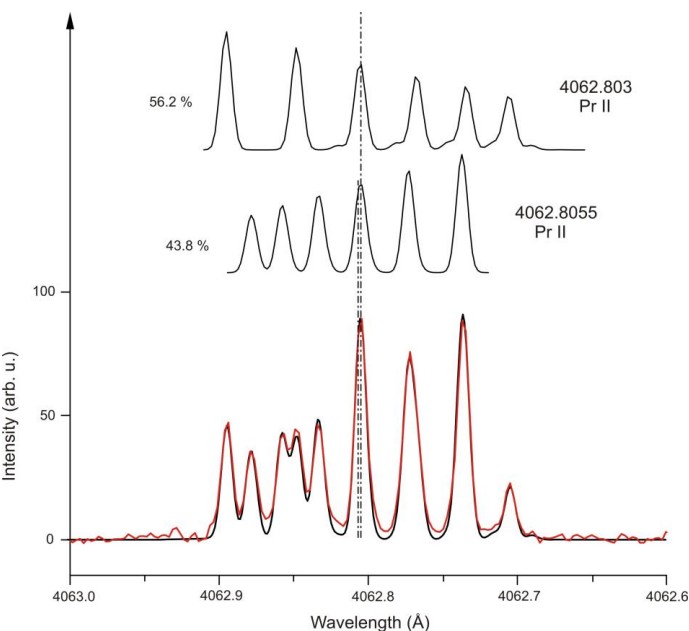

**Figure 8.** A blend situation of two Pr II lines with nearly the same cg wavelength. Red line: FT spectrum, black line: simulation, FWHM 1.6 GHz. In the upper part, normalized hf spectra of both lines are shown. Both profiles, added with the percentage given in the figure, gave the simulated sum profile (black), which describes the observed structure quite well. Thus, both cg wavelengths can be determined with high accuracy from the FT spectrum.

## 4. La II

Energy levels of La are also listed in Ref. [6]. Precise values of the hf structure constants of low lying metastable levels were obtained by a CLIBS technique by Höhle et al. [45] in 1982. These and other data were the basis of a theoretical interpretation of the hf structure of La II by Bauche et al. [46]. Later, CLIBS methods were also used by Li [47], Li [48], and Liang [49]. Some hf constants were determined by Lawler et al. [50] using an FT spectrum. Laser spectroscopic investigations of La II lines were made by Furmann et al. [51,52].

The FT spectra available in Graz (3225–16,600 Å) were taken in the group of Ferber (Laser Centre, University of Latvia, Riga, Latvia) with support of Kröger (Hochschule für Technik und Wirtschaft, Berlin, Germany). The near infrared part was analyzed in co-operation with the group of Basar (Physics Department, Istanbul University, Istanbul, Turkey). This was the first investigation of a La spectrum at wavelengths higher than 10,600 Å (investigated spectral range 8330–16,600 Å) [10]. The spectra were calibrated carefully using Ar II spectral lines. This allowed, together with the knowledge of the hf constants of the involved levels, a precise determination of transition wavelengths. From these values, improved energy values were determined. The results are submitted for publication [53]. Laser spectroscopy was performed in Graz mainly on atomic lines of La. These investigations are supported from the theoretical side by the group of Dembczyński (Institute of Materials Research and Quantum Engineering, Poznań University of Technology, Poznań, Poland).

As in the case of Pr, in the La spectrum blend situations are also observed quite frequently, despite the fact that the number of lines appearing in the spectra is lower (ca. 10,500). Figure 9 shows a typical blend situation.

Figure 10 shows an example for a La II line with widely split hf components. The line at 4151.957 Å is split into two groups of components. Since all components are well resolved, at such line, an independent determination of the hf constants of both combining levels is possible.

Additionally, Zeeman patterns of La II lines were investigated using a hollow cathode discharge in the presence of a magnetic field [54].

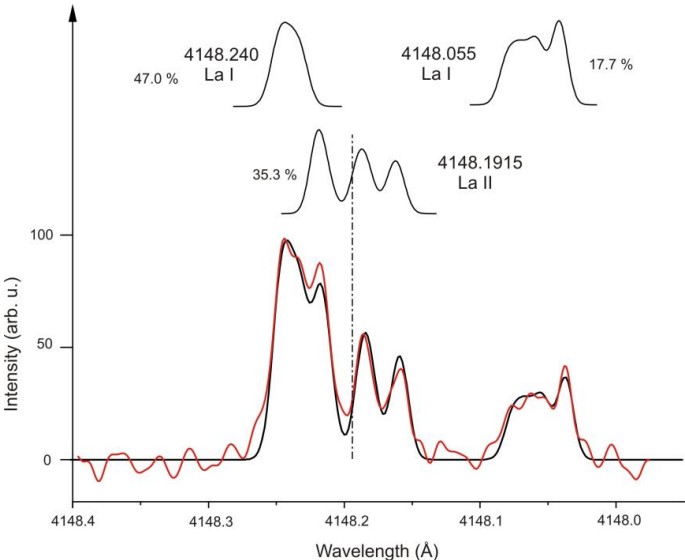

**Figure 9.** Blend of a La II and a La I line. Close to the blended lines another La I line appears. Red line: FT spectrum, black line: simulation, FWHM 2.4 GHz. Despite the fact that the signal-to noise ratio (SNR) is small (only 16 for the highest peak), the transitions can be clearly identified and their cg wavelength can be determined with good accuracy.

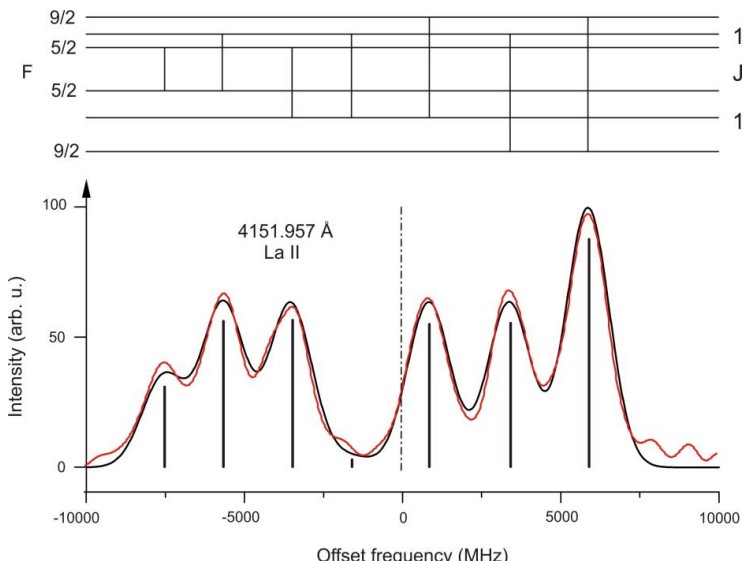

**Figure 10.** Example of a La II spectral line. Red line: FT spectrum, black line: simulation, FWHM 1.6 GHz. Also shown is the hf level scheme, the transitions and the components (theoretical intensity ratios).

## 5. Conclusions

Improved level energies of Ta II, Pr II, and La II were published. This paper is concerned with some peculiarities that had to be taken into account for an improvement of the energy values.

**Acknowledgments:** The author would like to thank all persons who contributed to the work described in the present article. Special thanks are devoted to G.H. Guthöhrlein, Universität der Bundeswehr Hamburg (Germany), for a very fruitful co-operation for more than 25 years. The work was theoretically supported by the group of J. Dembczyński, University of Technology, Poznań (Poland). Special thanks are devoted to W. Ernst, the present head (since 2002) of the Institute of Experimental Physics, Graz University of Technology, Graz, Austria, for allowing me to keep a room and a laser spectroscopy lab after my retirement (2014).

**Conflicts of Interest:** The author declares no conflict of interest.

# Appendix A

**Table A1.** Classification of the lines shown in the figures and data of the involved energy levels. Cols.: columns, Sp.: spectrum, tw: this work.

| Line | | | | Upper level | | | | | Lower Level | | | | | References to Cols. | | |
|---|---|---|---|---|---|---|---|---|---|---|---|---|---|---|---|---|
| Fig. No. | Sp. | Wavelength (Å) | SNR | Energy (cm$^{-1}$) | J | P | A (MHz) | B(MHz) | Energy (cm$^{-1}$) | J | P | A (MHz) | B(MHz) | 5,10 | 8,9 | 13,14 |
| 1 | 2 | 3 | 4 | 5 | 6 | 7 | 8 | 9 | 10 | 11 | 12 | 13 | 14 | 15 | 16 | 17 |
| 1 | Ta II | 3127.748 | 86 | 41708.994 | 5 | o | 914(5) | 350(100) | 9746.376 | 4 | e | 303(10) | 1680(300) | [25] | [25] | [25] |
| 1 | Ta I | 3127.864 | 30 | 31961.442 | 5/2 | o | 1243(3) | 740(10) | 0 | 3/2 | e | 509.084(1) | −1012.238(8) | [18] | tw | [55] |
| 2 | Ta II | 3487.803 | 21 | 54048.682 | 5 | o | 650(10) | 1200(200) | 25385.546 | 4 | e | 730(20) | 0(200) | [25] | [25] | [25] |
| 3 | Ta II | 3037.503 | 39 | 39743.636 | 4 | o | 955(5) | −1200(100) | 6831.437 | 3 | e | 360(10) | 970(200) | [25] | [25] | [25] |
| 3 | Ta II | 3037.532 | 17 | 46387.287 | 2 | o | 1802(15) | −130(50) | 13475.416 | 1 | e | −480(4) | 724(20) | [25] | [25] | [21] |
| 5 | Pr II | 5800.859 | 16 | 17676.112 | 5 | e | 805 | - | 442.060 | 5 | o | 1910.3(21) | - | [40] | [34] | [42] |
| 7 | Pr I | 5161.710 | 15 | 33932.700 | 13/2 | o | 737(1) | - | 14564.673 | 13/2 | e | 577(2) | - | [38] | [38] | [37] |
| 7 | Pr I | 5161.717 | 280 | 29899.954 | 17/2 | o | 529(1) | - | 10531.951 | 17/2 | e | 546(3) | - | [38] | [38] | [38] |
| 7 | Pr II | 5161.746 | 38 | 23261.402 | 5 | e | 581.9(3) | −19(5) | 3893.46 | 6 | o | 902.1 | - | [40] | [41] | [34] |
| 8 | Pr II | 4062.803 | 55 | 28009.828 | 7 | e | 556.5(7) | −34(25) | 3403.226 | 6 | o | −146.5(4) | - | [40] | [41] | [42] |
| 8 | Pr II | 4062.8055 | 34 | 27604.990 | 6 | e | 597.0(5) | 1(32) | 2998.412 | 7 | o | 1435.2(16) | - | [40] | [41] | [42] |
| 9 | La I | 4148.055 | 6 | 33820.316 | 1/2 | o | −232.7(60) | - | 9719.429 | 3/2 | e | −655.138 | −33.249 | tw | [56] | [57] |
| 9 | La II | 4148.1915 | 12 | 51524.005 | 2 | e | −220(3) | - | 27423.911 | 1 | o | 886.9(15) | −18.9(48) | [53] | [53] | [49] |
| 9 | La I | 4148.240 | 16 | 38903.885 | 5/2 | e | 181(3) | - | 14804.067 | 5/2 | o | 335.01(74 ) | 23.64(95) | [58] | [58] | [59] |
| 10 | La II | 4151.957 | 24 | 25973.360 | 1 | o | 547.3(30) | 27(7) | 1895.128 | 1 | e | −1128.1(0.9) | 49.8(65) | [53] | [52] | [45] |

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
