# Peer review of "The Role of the Hyperfine Structure for the Determination of Improved Level Energies of Ta II, Pr II and La II"

_atoms, doi:10.3390/atoms5010010_

Round 1

Reviewer 1 Report

The article  presents results of research performed by the author as a continuation of previous work on the same general subject. It is well written and provides some new results that complement existing published data. I recommend acceptance of this paper. 

The are are some minor corrections that could be made:

- The values in Table I seem to be from an old refernce (Atomic Data and Nuclear Data Tables 42, 189 (1991). A more recent compilation exists (ADNDT 90, 75 (1995)) presentig somewhat different values (at least for Ta).

- In the paper number references are given at the end of the phrases and not, as it is more usual, immediately after the authors' names.

- At line 217, instead of  "[39 and references therein]" it should be "[39] and references therein".

- In the list of references, some doi numbers are incorrect, starting with the first one. Usually there are missing dots. The authors should review the references carefully.

Author Response

I would like to thank the reviewers for their comments on the paper. I have considered all suggestions:

Reviewer 1:

- some figures in Table 1 are changed following the data given in the newer compilation of nuclear moments, suggested by the referee

- in connection with this point, reference [4] is changed

- ref. [39 and references therein]. is changed to ref. [39] and references therein.

- I checked all DOI addresses and changed the wrong addresses.

Reviewer 2 Report

The manuscript reviews recent results on the spectroscopy of emission lines of ions of Ta, Pr, and La in hollow-cathode discharges. The spectra of these ions are determined by a dense structure of multi-electron configurations and show a high density of lines, partly with overlapping hyperfine structure. The manuscript gives an overview over the experimental and analysis methods that are applicable to analyze these complex spectra and gives an overview over the recently published literature in this field. The manuscript is clearly written and gives useful guidelines also for the analysis of other spectra, for example of lanthanides or actinides. I recommend the manuscript for publication in the present form.

Author Response

I would like to thank the reviewers for their comments on the paper. I have considered all suggestions:

Reviewer 2:

- thanks for reading, there were no specific changes desired in the report